# Dietary Influence on Systolic and Diastolic Blood Pressure in the TwinsUK Cohort

**DOI:** 10.3390/nu12072130

**Published:** 2020-07-17

**Authors:** Panayiotis Louca, Olatz Mompeo, Emily R. Leeming, Sarah E. Berry, Massimo Mangino, Tim D. Spector, Sandosh Padmanabhan, Cristina Menni

**Affiliations:** 1Department of Twin Research, King’s College London, St Thomas’ Hospital Campus, Westminster Bridge Road, London SE1 7EH, UK; panayiotis.louca@kcl.ac.uk (P.L.); olatz.mompeo@kcl.ac.uk (O.M.); emily.leeming@kcl.ac.uk (E.R.L.); massimo.mangino@kcl.ac.uk (M.M.); tim.spector@kcl.ac.uk (T.D.S.); 2Department of Nutritional Sciences, King’s College London, London WC2R 2LS, UK; sarah.e.berry@kcl.ac.uk; 3NIHR Biomedical Research Centre at Guy’s and St Thomas’ Foundation Trust, London SE1 9RT, UK; 4Institute of Cardiovascular & Medical Sciences, University of Glasgow, Glasgow G12 8QQ, UK; Sandosh.Padmanabhan@glasgow.ac.uk

**Keywords:** nutrients, hypertension, blood pressure, management, prevention, diet

## Abstract

Nutrition plays a key role in blood pressure (BP) regulation. Here, we examine associations between nutrient intakes and BP in a large predominantly female population-based cohort. We assessed the correlation between 45 nutrients (from food frequency questionnaires) and systolic BP/diastolic BP (SBP/DBP) in 3889 individuals from TwinsUK not on hypertensive treatments and replicated in an independent subset of monozygotic twins discordant for nutrient intake (17–242 pairs). Results from both analyses were meta-analysed. For significant nutrients, we calculated heritability using structural equation modelling. We identified and replicated 15 nutrients associated with SBP, 9 also being associated with DBP, adjusting for covariates and multiple testing. 14 of those had a heritable component (*h*^2^: 27.1–57.6%). Strong associations with SBP were observed for riboflavin (Beta(SE) = −1.49(0.38), *P* = 1.00 × 10^−4^) and tryptophan (−0.31(0.01), *P* = 5 × 10^−4^), while with DBP for alcohol (0.05(0.07), *P* = 1.00 × 10^−4^) and lactose (−0.05(0.0), *P* = 1.3 × 10^−3^). Two multivariable nutrient scores, combining independently SBP/DBP-associated nutrients, explained 22% of the variance in SBP and 13.6% of the variance in DBP. Moreover, bivariate heritability analysis suggested that nutrients and BP share some genetic influences. We confirm current understanding and extend the panel of dietary nutrients implicated in BP regulation underscoring the value of nutrient focused dietary research in preventing and managing hypertension.

## 1. Introduction

Hypertension is the most prevalent modifiable risk factor for cardiovascular (CVD) morbidity and mortality [1]: for every 10 mmHg reduction in systolic blood pressure (SBP), the risk decreases by 11% [2,3]. Yet, hypertension prevalence is mounting, with a 1.5 fold increase estimated by the year 2025, affecting 1.5 billion individuals [4]. Blood pressure (BP) is determined by complex interactions between genetics and environmental exposures [5]. Some environmental factors are non-modifiable, such as age, ethnicity and gender [6]. However, others are modifiable, with extensive evidence supporting lifestyle modification, including physical activity, smoking and particularly dietary changes as efficacious first-line therapies for hypertension [7]. Studies have reported that adhering to a Med/DASH diet improves CVD outcomes [8], and recently there has been a push to move research towards whole dietary patterns [9]. However, due to the complexity of dietary patterns, many important nutrient effects may be overlooked.

Numerous studies have evidenced the relative effects of single nutrients on BP, including salt, potassium and alcohol. For instance, Cochrane and collaborators found that a 4.4 g/day reduction in salt (1733 mg sodium) reduced BP by 4.18/2.06 mmHg [10], which was significantly higher in hypertensive individuals (5.39/2.82 mmHg) compared to normotensives (2.42/1 mmHg) [10]. Additionally, a meta-analysis of 29 randomised clinical trials (RCTs) showed that an increase in potassium of ≥20 mg/d led to a BP reduction of 4.9/2.7 mmHg, including trials with hypertensive and normotensive subjects [11]. These results highlight the value of nutrient based research to control BP.

In the present study, we assess the role of 45 nutrient intakes estimated from Food Frequency questionnaires on SBP and diastolic BP (DBP) in a large cohort of twins. Having identified nutrient intakes associated with SBP or DBP, we validated the results using identical twins discordant for that particular nutrient. This allowed us to isolate the non-genetic contribution of nutrient intakes upon blood pressure. Finally, given the twin nature of our data, we estimated heritability of the associated nutrient intakes.

## 2. Methods

### 2.1. Study Population

Study participants were twins enrolled in the TwinsUK registry, a national register of adult twins recruited as volunteers without selecting for particular disease or traits [12]. We included 3889 predominantly female twins that completed a 131-item validated food frequency questionnaire (FFQ) between 1996 and 2015 [13] and had a concurrent BP measurement (within 0.16(SD = 0.29) years).

Twins provided informed written consent and the study was approved by St. Thomas’ Hospital Research Ethics Committee (REC Ref: EC04/015). Data relevant to the present study include SBP/DBP, BMI and zygosity (determined by methods previously outlined [14]).

#### 2.1.1. Assessment of Blood Pressure

Clinic BP was measured by a trained nurse using either the Marshall mb02, the Omron Mx3 or the Omron HEM713C Digital Blood Pressure Monitor performed with the patient in the sitting position for at least 3 min. At each visit, the cuff was placed on the subject’s arm so that it was approximately 2–3 cm above the elbow joint of the inner arm, with the air tube lying over the brachial artery. The subject’s arm was placed on the table or supported with the palm facing upwards, so that the tab of the cuff was placed at the same level of the heart. Triplicate measurements were taken with an interval of approximately 1 min between each reading, with mean of second and third measurements recorded.

#### 2.1.2. Nutrient Data

Intakes of 46 nutrients were estimated from a validated 131-item food frequency questionnaire (FFQ) based upon the EPIC FFQ [13]. Prior to analysis intake frequencies were adjusted for total energy intake using the residual method [15]. FFQs were then coded and processed using FETA [16], an open-source, cross-platform tool designed to process dietary data from the EPIC FFQ, in accordance with their guidelines. The default nutritional database of which is based on the McCance and Widdowson’s The Composition of Foods (5th edition) [17].

#### 2.1.3. Dietary Indices

To determine the effects of the whole diet, we employed the most prominently used dietary indices including, the NOVA classification system [18], the Healthy Eating Index (HEI) [19], the alternate Healthy Eating Index (aHEI) [20], the Dietary Approaches to Stop Hypertension (DASH) score [21], the Alternate Mediterranean Diet Score (aMED) [22], the Dietary Quality Index International (DQI-I) [23], the Plant Diversity Index (PDI) [24], the Healthy PDI (hPDI) [24] and the Unhealthy PDI (uPD) [24]. Descriptions of the indices can be found in Appendix A.

### 2.2. Statistical Analysis

Statistical analysis was performed using R version 3.6.2.

Linear mixed models were used to investigate the associations of each nutrient with SBP and DBP in the discovery sample (excluding monozygotic (MZ) twins with nutrient intake over one SD apart). Analyses were adjusted for age, gender, BMI, family relatedness and multiple testing using Benjamini–Hochberg correction (FDR < 0.05).

The MZ discordant twin pairs were then used to replicate the significant findings from the discovery group. Associations that passed the 5% level of significance or were in the same direction as the discovery group were considered replicated. Finally, we combined the results of both analyses using an inverse variance fixed effect meta-analysis.

A backwards stepwise regression, including the BP-associated nutrients, was then employed in the overall sample to identify nutrient intakes independently associated with SBP and DBP (p < 0.05). Nutrients independently associated with SBP and DBP were then linearly combined into an SBP and DBP nutrient score, respectively.

We further investigated the role of dietary indices on BP using linear mixed model adjusting for age, sex, BMI, family relatedness and multiple testing in order to look at the effect of diet as a whole. For the NOVA system, subjects were stratified into tertiles of intakes for each level of processing, to determine disparities of associations between strata using the same linear mixed models.

#### Heritability

Taking advantage of the twin nature of our data, we estimated heritability of nutrient intakes and BP using structural equation modelling to decompose the observed phenotypic variance into three latent sources of variation: additive genetic variance (A), shared/common environmental variance (C) and non-shared/unique environmental variance (E) [25]. Additive genetic influences are indicated when monozygotic (MZ) twins are more similar than dizygotic (DZ) twins. Heritability is defined as the proportion of the phenotypic variation attributable to genetic factors, and is given by the equation, *h*^2^ = (A)/(A + C + E). The Akaike information criterion (AIC) was used to determine the best-fitting model (among ACE, AE and CE models). The model with the lowest AIC reflects the best balance of goodness of fit and parsimony [25]. The maximum likelihood method of model fitting was applied to the raw data using the R package MET.

We further investigated whether SBP/DBP share underlying genetics factors with the SBP/DBP nutrient scores. We used bivariate genetic model employing Cholesky decomposition and psychometric common pathway model [26,27].

## 3. Results

The demographic characteristics of the study population are presented in Table 1. A total of 3889 individuals from TwinsUK with BP measures and estimated nutrient intakes were included in the analysis. Of these, 1326 were MZ pairs, 1946 DZ pairs and 617 singletons. The study sample was predominantly female (97.9%), had mean age 54.9(12.8) years, and slightly overweight (BMI = 25.32(4.41) Kg/m^2^). See Table 1.

### 3.1. Nutrient Intake-Blood Pressure Associations

In the discovery cohort, out of the 45 nutrients, we found 17 nutrients associated with SBP, and 10 of those also associated with DBP after adjusting for age, sex, BMI, total energy intake, family relatedness and multiple testing using the Benjamini–Hochberg correction (FDR < 0.05) (see Appendix A).

We then validated our results in the MZ discordant groups, after identifying between 17 and 242 twin pairs discordant for nutrient intakes. A total of 15 nutrients were associated with SBP and 9 also with DBP in the MZ discordant group (Figure 1). We further combined the results from the discovery and replication datasets using inverse variance fixed effect meta-analysis and found that all successfully replicated nutrient-BP associations were significant after meta-analysis (FDR < 0.05) (Figure 1).

To identify nutrients independently associated with BP, we linearly combined the significantly associated nutrients in the whole population by running a backwards stepwise regression including age, BMI, sex and family structure. Six nutrients were independently associated with SBP. These include alcohol, water, saturated fatty acid, riboflavin, tryptophan and biotin, together explaining 22.2% of the variance.
(1)$$\begin{array}{cc} {\mathrm{SBP}}_{\mathrm{score}}= & 179.072827+\left(0.0716177\times \mathrm{alcohol}\right)-\left(0.0007501\times \mathrm{water}\right)\\ & -\left(0.0968079\times \mathrm{saturated}\;\mathrm{fats}\right)+\left(0.0702823\times \mathrm{riboflavin}\right)\\ & -\left(0.0150996\times \mathrm{tryptophan}\right)-\left(0.1110900\times \mathrm{biotin}\right) \end{array}$$

Three nutrients were independently associated with DBP, explaining 13.6% of variance, they include alcohol, carbohydrates and biotin:(2)$${\mathrm{DBP}}_{\mathrm{score}}=49.956227+\left(0.070326\times \mathrm{alcohol}\right)+\left(0.009522\times \mathrm{carbohydrates}\right)-\left(0.047617\times \mathrm{biotin}\right)$$

### 3.2. Dietary Indices and Blood Pressure

We estimated the influence of nine dietary indices and both SBP/DBP and observed no associations between any of the dietary indices and neither SBP nor DBP (Appendix A). No significant associations were found also when we stratified the population by intake using the NOVA classification system.

### 3.3. Heritability

We estimated heritability of the 15 BP-associated nutrient using structural equation modelling and found that the best fitting model for 14 of those was the AE model with heritability estimates ranging from 27.1% [21.3%; 33%] for tryptophan to 52.7% [48.1%; 57.3%] for carbohydrates (Figure 2; Appendix A). Iodine, on the other hand, appeared to be only environmentally determined.

The best fitting model for SBP/DBP scores was the AE model, with heritability estimates of 54% [49%; 58%] and 58% [53%; 62%], respectively (Appendix A).

Previous studies by us and others have found SBP/DBP to have strong heritable component [28]. In this data, in line with previous findings, the best fitting model for both SBP and DBP was the AE model with heritability estimates of 53.7% for SBP and 57.6% for DBP.

We further investigated how much of the BP heritability is common to nutrient intakes (estimated with the SBP and DBP dietary scores) and found a shared heritability of 31.6% for SBP and of 30% for DBP.

## 4. Discussion

In one of the most comprehensive studies, incorporating 45 nutrients, 9 dietary indices and heritability to investigate diet-BP associations, we identified and replicated 15 nutrients to be associated with SBP and 9 with DBP. We also generated a nutrient score for both SBP and DBP from independently associated nutrients that, respectively, explained 22.2% and 13.6% of the variance in SBP/DBP. Both of which were positively associated with SBP and DBP respectively. Furthermore, we found that 14 out of the 15 unique nutrients were genetically determined, with heritability ranging from 27.1% to 52.7%. This is consistent with previous reports on macro- and micro-nutrients heritability ranging from 21 to 55% [29,30].

Additionally, in line with our previous results [28], we find that both SBP and DBP are heritable (h^2^ = 53.7% and 57.6% respectively). Here we also report that BP and nutrients share 31.6% and 30% of the genetic influence.

The BP associated nutrients included a mix of macronutrients, amino-acids, vitamins and minerals highlighting the need to look beyond single nutrients when exploring the impact of diet on BP. The largest beneficial effects were observed for B vitamins, riboflavin (vitamin B2) for SBP and biotin (vitamin B7) for DBP.

### Nutrients Independently Associated with BP

We identified six nutrients independently associated with SBP; this included alcohol, water, saturated fatty acids, riboflavin, tryptophan and biotin, which explained 22.2% of the variance. Three of those nutrients were also independently associated with DBP; these were, alcohol, carbohydrates and biotin, explaining 13.6% of variance. The percentage of variance explained by nutrients is much higher than that explained for instance by genetic factors. Indeed, over 1477 common single nucleotide polymorphisms associated with BP explain 5.7% of population phenotypic variance in SBP [5,31].

Of the six independently associated nutrient intakes, alcohol, tryptophan and riboflavin elicited large effects, in line with previously reported literature [32,33,34,35,36,37,38,39,40].

RIBOFLAVIN: Riboflavin, also known as vitamin B₂, is a water-soluble vitamin found in food, predominantly milk and egg products [41], and also used as a dietary supplement. The effect of riboflavin in reducing BP has been previously reported [32]. Riboflavin acts on BP in a gene-nutrient interaction involving the gene encoding methylenetetrahydrofolate reductase [33], potentially stabilising variants within this gene to restore 5-methyltetrahydrofolate concentrations to improve nitric oxide bioavailability, a potent vasodilator in BP control [32].

In our sample, riboflavin intake was above that of the average UK intake [42] (2.38 mg and 1.59 mg respectively). Here we find that the effect of riboflavin on SBP is independent from that of the other nutrients, highlighting the value of riboflavin intake for blood pressure control. Moreover, we report that riboflavin is moderately heritable, suggesting more than a third of the observed individual differences in riboflavin intake we observed, may be attributable to genetic individual differences, while the remaining 65% is due to the environment.

ALCOHOL: The present findings also illustrated that alcohol exerted the greatest deleterious effect upon both SBP and DBP, despite the average alcohol intake of our sample (Table 1) being below the average UK intake (9.54 g/d and 12.4 g/day, respectively) [43]. This is ubiquitous with that of numerous other studies [34,35,36,37,44]. Pajak and colleagues reported a strong effect for alcohol consumption and both SBP and DBP, where even low volume and frequency exerted a deleterious effect [36]. Literature repeatedly reports a J-shaped association between alcohol intake and CVD outcomes [45], but disaggregation suggests a linear-association with SBP [45]. This linear dose-response relationship was reported to exert the strongest effect in females [37]. This increased susceptibility of alcohol upon blood pressure in females may have been an observation within our female dominant cohort.

TRYPTOPHAN: Tryptophan, an essential amino acid, is commonly derived from meat products and soybeans [46]. Tryptophan is a precursor of serotonin synthesis and shown to reduce BP [47]. Serotonin is a monoaminergic neurotransmitter influencing vasoconstriction, yet the exact mechanisms underlying the relationship between tryptophan, serotonin and reduced BP are not known [48].

The association between tryptophan and BP is controversial, with some studies reporting negative correlations [49], and others, as in our study, identifying associations between tryptophan levels and decreased SBP [38,39,40]. This is speculated to be relating to differing dietary sources (animal versus plant) [50]. Moreover, tryptophan-containing peptides derived from the enzymatic hydrolysis of dietary protein are thought to interfere with the renin-angiotensin axis by inhibiting the rate-limiting, angiotensin converting enzyme, thereby mitigating BP [50].

These inconsistent results may also be the result of inter-individuality in sympathetic nervous system activity [51], which is the proposed mechanism in which tryptophan attenuates BP [52].

Here we report a strong negative correlation between tryptophan and SBP, supporting the role of tryptophan in BP control. Tryptophan is naturally available from animal and plant proteins [53], but increasing the consumption of a single amino-acid naturally is unfeasible. Suggesting that administration and continuous treatment of tryptophan may improve BP [48]. As tryptophan seems safe to consume, the potential health benefits has led to plant molecular genetic engineering endeavours to generate high tryptophan cereals and legumes [46].

LACTOSE: We found lactose intake to be associated to lower DBP in the univariate analysis, though lactose was not independently associated with DBP in the multivariate model. This is in line with previous studies reporting dairy products to have a lowering effect on BP [54] and lactose to have a stronger effect compared to calcium or phosphorus [54].

We also note that we did not observe any association between SBP/DBP and sodium intake [55]. This is probably due to limitations of the FFQ in estimating true sodium intake [56].

Interestingly, we observed no association between any of the dietary indices, while we found strong associations with single nutrients, highlighting the complexity of dietary patterns and underscoring how dietary indices may overlook some minor effects elicited by nutrients [9]. Dietary indices typically define dietary components that are considered important for the goal of that index, thence limiting the utility of that index [57]. Ultra-processed foods have received much attention recently in relation to health and disease [58]. In our study, when stratifying the population based upon the NOVA classification system, we detected no major differences between associations with BP in any of the NOVA strata. This novel finding is counterintuitive to our current beliefs on processed foods [18], reiterating the specificity of the index, whereby nutritional value is overlooked.

Although nutritional research is now moving away from a single nutrient approach and is focused on studying whole foods and dietary patterns, the present study underscores the value of nutrient focused dietary research in preventing and managing hypertension and our study is strengthened by numerous factors. Firstly, the large sample size facilitates the comprehensive analysis of numerous nutrients and retains sufficient sample size in the MZ replication groups. Secondly, the co-twin control study design results in the replication samples being matched for both measured and unmeasured factors, strengthening causal inferences [59]. Additionally, the use of co-twins provides the most matched genetic controls feasible [60], facilitating detrimental environmental agents of BP to be explored. Thirdly, BP was measured by experienced research nurses, whereby the first measurement was discarded and an average of two remaining measurements was used, strengthening the accuracy and precision of measurements.

We also noted some study limitations. First, findings were based upon estimated intakes generated from FFQ, limiting reliability and reproducibility as these are prone to reporting bias and random error [61,62]. Second, our study sample is predominantly female, so results may not be generalizable to men. Third, we are unable to infer causality from this cross-sectional research, requiring future interventions. This future work should investigate the mechanisms by which nutrients influence blood pressure pathways and the inter-individual differences in physiological responses to nutrients with clinical trials.

Our findings confirm current understanding, highlights the utility of nutrient focussed research to control BP and directs attention to the potential use of nutrient supplemented foods for BP control.

## Figures and Tables

**Figure 1 nutrients-12-02130-f001:**
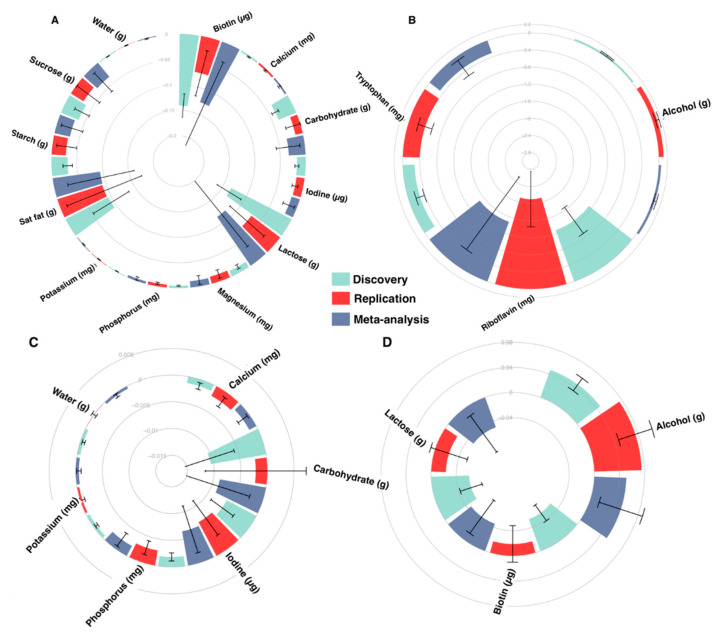
Meta-analysed nutrient associations with blood pressure. Nutrient-BP associations from the discovery population (False Discovery Rate (FDR) < 0.05), replication population (P < 0.05 or same direction beta) and meta-analysis. Because of the scale and comparative differences of effects, panels display varying effect sizes to facilitate visualisation of effects. SBP associations are illustrated in panels (**A**) (Beta: −0.2 to 0) and (**B**) (Beta: −2.8 to 0.2), and DBP associations are illustrated in panel (**C**) (Beta: −0.015 to 0.005) and (**D**) (Beta: −0.04 to 0.08). Error bars display SE. Results from the discovery cohort are represented in teal, from the replication cohort in red and from meta-analyses in blue.

**Figure 2 nutrients-12-02130-f002:**
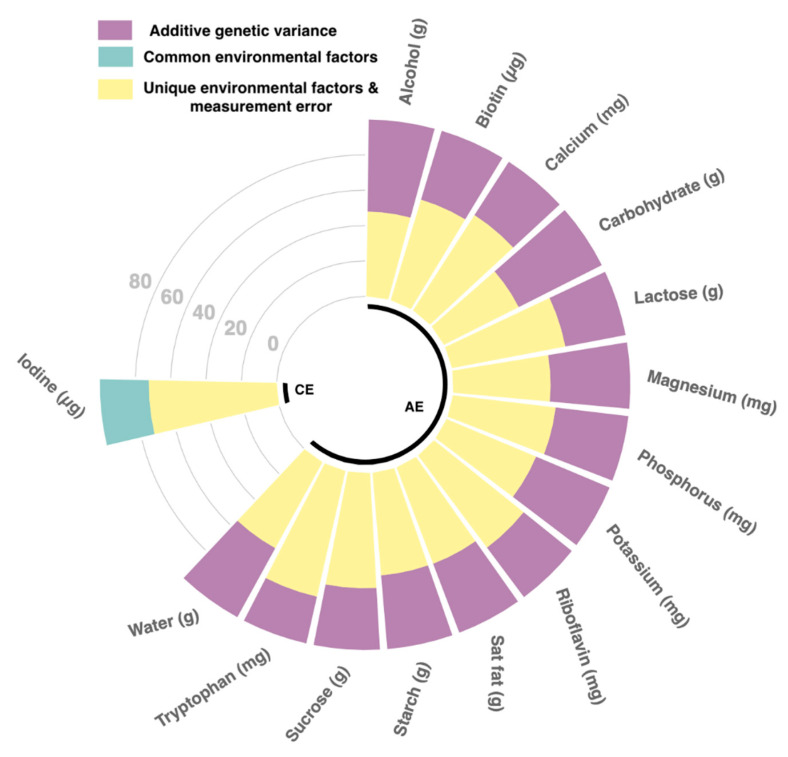
Circus plot of heritability for significantly associated nutrients. Plot of heritability analysis depicting sources of phenotypic variation for the 15 nutrients significantly associated with blood pressure. Nutrients in group AE were genetically derived, whereas nutrients in group CE were environmentally determined (model lowest AIC). Purple bars represent additive genetic variance, teal represent common environmental factors and yellow specific environmental factors & error.

**Table 1 nutrients-12-02130-t001:** Characteristics of the study population (*n* = 3889).

Phenotype	*n*	%	
*N*	3889		
Female, *n* (%)	3808	97.2	
MZ pairs	1326	34.1	
	**Mean**	**SD**	
Age, yrs.	54.9	12.8	
BMI, kg/m^2^	25.3	4.4	
SBP, mmHg	121.3	16.1	
DBP, mmHg	76	10.5	
**Nutrients *****	**Mean**	**SD**	**% energy**
Water, g	2719.24	780.97	
Alcohol, g	9.54	13.56	4.43
Carbohydrates, g	247.97	77.8	51.16
Starch, g	121.49 g	44.09	25.07
Total sugars, g	123.82	45.92	25.55
Sucrose, g	48.56	22.1	10.02
Maltose, g	3.8	2.27	0.58
Lactose, g	20.27	10.6	4.18
Saturated fats, g	25.47	10.43	11.82
Potassium, mg	3920	1058.97	
Calcium, mg	1096.2	386.02	
Magnesium, mg	340.48	96.4	
Phosphorus, mg	1494.38	421.83	
Iodine, µg	214.46	82.34	
Riboflavin, mg	2.38	0.88	
Tryptophan, mg	17.06	4.76	
Biotin, µg	47.39	14.7	

BMI, body mass index; SBP, systolic blood pressure; DBP, diastolic blood pressure. * Only nutrients associated with SBP and DBP after adjusting for covariates and multiple testing are included in Table 1. The full list of the nutrients included is provided in Appendix A.

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
