# Peer review of "Dietary Influence on Systolic and Diastolic Blood Pressure in the TwinsUK Cohort"

_nutrients, 2020, doi:10.3390/nu12072130_

Round 1
Reviewer 1 Report
The authors presented an interesting study investigating a wide range of nutrients in their association with brachial systolic and diastolic blood pressure (BP). I have the following comments:
- Out of the 15 nutrients found to have a significant association with BP, only alcohol had positive association, meaning it was the only nutrient that would be detrimental to BP. Was this surprising? Furthermore, I found it intriguing that sodium didn't come out in the association, given the surmounting evidence that link salt with BP. Perhaps the authors could elaborate on this further?
-
It was unclear to me what SBPscore and DBPscore represent. Is it higher the better or lower the better? With this equation, assuming a higher score is worse, alcohol and riboflavin are the only two nutrients that would increase SBP/DBPscore. Are these scores synonymous with SBP and DBP, respectively? Furthermore, it appears that the association of riboflavin changed directions (negative association in Table S2, but additive to SBPscore and DBPscore). However, it was stated in the discussion that riboflavin reduces BP, thus I would have expected a negative coefficient for riboflavin.
- The variance in SBP/DBP explained by the nutrients is actually quite high. Could the authors perhaps include in the discussion how this percentage compares to other factors that influence BP?
Other minor comments:
- pg 3, line 100: does FDR stand for false discovery rate, or is it an adjusted P value with Benjamini-Hochberg correction? It may be helpful to clarify, as false discovery rate has a different meaning in this procedure. It appears that a false discovery rate of 0.1 was used in the calculation of the Benjamini-Hochberg critical value.
- pg 1, line 42, "effecting" should be "affecting"
- pg 3, line 98, MZ was not previously defined
- more detailed results for the heritability would be helpful, at least in supplementary format; e.g. a table of the estimates, as well as the confidence intervals of the heritability estimates in the manuscript text
- pg 4, line 231-232, sentence seems incomplete
Author Response
- Out of the 15 nutrients found to have a significant association with BP, only alcohol had positive association, meaning it was the only nutrient that would be detrimental to BP. Was this surprising? Furthermore, I found it intriguing that sodium didn't come out in the association, given the surmounting evidence that link salt with BP. Perhaps the authors could elaborate on this further?
Authors’ response: We thank the reviewer for this comment. The lack of association with sodium is likely due to the limitations of FFQ in estimating true sodium intake (Kelly, Nutr Metab Cardiovasc Dis, 2015).
A difference in sodium intake of 6 gm/day could be associated with average differences in SBP of 5mmHg at age 15-19 yrs and 10mmHg at age 60-69 years (Law, BMJ, 1991).
Regarding alcohol being the only nutrient to be detrimental to BP, we were not surprised as alcohol and sodium are the two main nutrients known to increase BP and sodium from FFQ is underestimated. Furthermore, alcohol has been shown to increase BP and a causal role for alcohol has been shown in mendelian randomisation studies (Brien, BMJ, 2011).
- It was unclear to me what SBPscore and DBPscore represent. Is it higher the better or lower the better? With this equation, assuming a higher score is worse, alcohol and riboflavin are the only two nutrients that would increase SBP/DBPscore. Are these scores synonymous with SBP and DBP, respectively? Furthermore, it appears that the association of riboflavin changed directions (negative association in Table S2, but additive to SBPscore and DBPscore). However, it was stated in the discussion that riboflavin reduces BP, thus I would have expected a negative coefficient for riboflavin.
Authors’ response: We apologise with the reviewer for the lack of clarity. We have now specified that the two scores are a linear combination of nutrients independently associated with SBP and DBP respectively and that the higher the score, the higher the BP.
Regarding, riboflavin, the reviewer is correct. The beta is negative in the univariate analysis in both the discovery and replication cohorts (Table S2), and the coefficient is positive in the multivariable model. The reason for this is the collinearity between nutrients in the backwards stepwise regression model.
- The variance in SBP/DBP explained by the nutrients is actually quite high. Could the authors perhaps include in the discussion how this percentage compares to other factors that influence BP?
Authors’ response: Following the reviewer comment, we have now added as a comparison that the percentage of variance explained for instance by genetic factors (900 SNPs) is 5.7%. (Evangelou, Nature Gen, 2018) .
Line 220: The percentage of variance explained by nutrients is much higher than that explained for instance by genetic factors. Indeed 1477 common single nucleotide polymorphisms associated with BP, explain 5.7% of population phenotypic variance in SBP [5,31].
Other minor comments:
pg 3, line 100: does FDR stand for false discovery rate, or is it an adjusted P value with Benjamini-Hochberg correction? It may be helpful to clarify, as false discovery rate has a different meaning in this procedure. It appears that a false discovery rate of 0.1 was used in the calculation of the Benjamini-Hochberg critical value.
Authors’ response: We clarified that a Benjamin Hochberg correction was used in the discovery cohort, taking values with an FDR <0.05 as significant.
pg 1, line 42, "effecting" should be "affecting"
Authors’ response: Thank you for identifying this error. We have rectified this as per.
pg 3, line 98, MZ was not previously defined
Authors’ response: We have now specified that MZ stands for monozygotic.
more detailed results for the heritability would be helpful, at least in supplementary format; e.g. a table of the estimates, as well as the confidence intervals of the heritability estimates in the manuscript text
Authors’ response: Following the reviewer’ suggestion, we have included CIs and provided all the estimates in Supplementary Tables 5 and 6.
pg 4, line 231-232, sentence seems incomplete
Authors’ response: We have now rephrased the sentence.
Reviewer 2 Report
The work by Panayiotis Louca, Olatz Mompeo Masachs , Emily R Leeming , Sarah E Berry, Massimo Mangino, Tim D Spector , Sandosh Padmanabhan, and Cristina Menni is interesting. This study collected the data from a large number of human subjects (> 3,000 people), which may provide more insight into hypertension and nutrition.
My minor comments are as follows:
1. A higher resolution of Fig. 1 should be provided.
2. Since lactose was found to have the strongest association with BP, it would be great if you can touch on lactose-BP in the discussion part.
3. It is interesting in the tryptophan part. It will be more informative if the discussion is extended to the effects of tryptophan-containing peptides on BP. The current study may explain previous findings of BP-lowering effects of small tryptophan-containing peptides derived from foods such as fish and plant hydrolysates.
4. In the alcohol part, it may be more informative is the discussion about J- or U-shaped curve of alcohol and BP are added.
Author Response
A higher resolution of Fig. 1 should be provided.
Authors’ response: We have now provided higher resolutions figures with larger font for labels providing greater clarity.
Since lactose was found to have the strongest association with BP, it would be great if you can touch on lactose-BP in the discussion part.
Authors’ response: We thank you for this comment. We have now included the following in the discussion:
We found lactose intake to be associated to lower DBP in the univariate analysis, though lactose was not independently associated with DBP in the multivariate model. This is in line with previous studies reporting dairy products to have a lowering effect on BP and lactose to have a stronger effect compared more so than calcium or phosphorus [54].
It is interesting in the tryptophan part. It will be more informative if the discussion is extended to the effects of tryptophan-containing peptides on BP. The current study may explain previous findings of BP-lowering effects of small tryptophan-containing peptides derived from foods such as fish and plant hydrolysates.
Authors’ response: We thank the reviewer for this suggestion. We have now added the following to the discussion:
Moreover, tryptophan-containing peptides derived from the enzymatic hydrolysis of dietary protein are thought to interfere with the renin-angiotensin axis by inhibiting the rate-limiting, angiotensin converting enzyme, thereby mitigating BP [50].
In the alcohol part, it may be more informative is the discussion about J- or U-shaped curve of alcohol and BP are added.
Authors’ response: Following the reviewer’ suggestion we have now discussed about J or U-shaped curve of alcohol and BP.
In the discussion:
Literature repeatedly reports a J-shaped association between alcohol intake and CVD outcomes, but disaggregation suggests a linear-association with SBP [45].
Reviewer 3 Report
This study would likely be of significant interest as it describes modifiable factors associated with blood pressure.
Twin studies are an excellent way to tease out genetic and non-genetic influences and this study uses a large twin data set. It also examined the effects of both individual nutrients and dietary patterns on BP along with their heritability.
Some minor comments:
Line 42: affecting not effecting
Lines 107 & 185: Authors should clarify the term ‘nutrient score’ and how it was obtained
The Abstract should note that the population was almost all female as this is an important feature of the study group, and assumptions should not be made that it is representative of both sexes.
Paragraph starting at line 210: Be consistent with either riboflavin or B2
Figure 1. Colours should be written eg. meta-analysed associations BLUE
Author Response
Twin studies are an excellent way to tease out genetic and non-genetic influences and this study uses a large twin data set. It also examined the effects of both individual nutrients and dietary patterns on BP along with their heritability.
Authors’ response: We thank the reviewer for this positive evaluation of our manuscript.
Some minor comments:
Line 42: affecting not effecting
Authors’ response: We thank the reviewer for spotting this typo.
Lines 107 & 185: Authors should clarify the term ‘nutrient score’ and how it was obtained
Authors’ response: We thank you the reviewer for this comment. We have now clarified what is the nutrient score and how it was obtained.
The Abstract should note that the population was almost all female as this is an important feature of the study group, and assumptions should not be made that it is representative of both sexes.
Authors’ response: We have now specified that the study population was predominantly female.
Paragraph starting at line 210: Be consistent with either riboflavin or B2
Authors’ response: Following the reviewer’ suggestion we now consistently refer to riboflavin/B2 as riboflavin.
Figure 1. Colours should be written eg. meta-analysed associations BLUE
Authors’ response: We have now specified the colour coding in the legend as well.
Round 2
Reviewer 1 Report
Thank you for the clarification and response to comments. I have nothing further to add.